# Cardiolipin Regulates Mitochondrial Ultrastructure and Function in Mammalian Cells

**DOI:** 10.3390/genes13101889

**Published:** 2022-10-18

**Authors:** Zhitong Jiang, Tao Shen, Helen Huynh, Xi Fang, Zhen Han, Kunfu Ouyang

**Affiliations:** 1Department of Cardiovascular Surgery, Peking University Shenzhen Hospital, Shenzhen 518055, China; 2Department of Medicine, University of California San Diego, 9500 Gilman Drive, La Jolla, San Diego, CA 92093, USA

**Keywords:** cardiolipin, mitochondria, mouse models, mitochondrial function

## Abstract

Cardiolipin (CL) is a unique, tetra-acylated diphosphatidylglycerol lipid that mainly localizes in the inner mitochondria membrane (IMM) in mammalian cells and plays a central role in regulating mitochondrial architecture and functioning. A deficiency of CL biosynthesis and remodeling perturbs mitochondrial functioning and ultrastructure. Clinical and experimental studies on human patients and animal models have also provided compelling evidence that an abnormal CL content, acyl chain composition, localization, and level of oxidation may be directly linked to multiple diseases, including cardiomyopathy, neuronal dysfunction, immune cell defects, and metabolic disorders. The central role of CL in regulating the pathogenesis and progression of these diseases has attracted increasing attention in recent years. In this review, we focus on the advances in our understanding of the physiological roles of CL biosynthesis and remodeling from human patients and mouse models, and we provide an overview of the potential mechanism by which CL regulates the mitochondrial architecture and functioning.

## 1. Introduction

In mammals, mitochondria are involved in numerous metabolic and bioenergetic processes that are critical for cell survival. Mitochondria are considered as the powerhouse of the cell, generating the energy required for cellular metabolism through the process of oxidative phosphorylation (OXPHOS) [1]. These organelles are also involved in many other physiological processes in cells, such as programmed cell death, autophagy, redox signaling, and Ca^2+^ homeostasis [2,3,4]. Mitochondria contain two membranes, the outer mitochondrial membrane (OMM) and the inner mitochondrial membrane (IMM), and the compartment separated by these two membranes is called the intermembrane space (IMS). The cristae structure is the tubular invagination of the IMM that protrudes into the matrix and harbors an enzymatic complex involved OXPHOS. Phospholipids are the major components of the mitochondrial membrane bilayer. The OMM and IMM exhibit different lipid compositions and distributions of mainly phosphatidylcholine, phosphatidylethanolamine, phosphatidylinositol, phosphatidylserine, phosphatidic acid, and cardiolipin (CL) [5]. These compounds, with the exception of CL, are mainly synthesized in the endoplasmic reticulum (ER) and then transferred to the mitochondria. Phospholipids play critical roles in the structure, function, dynamics, and protein transport of mitochondria [5]. Changes in the phospholipid composition can affect the mitochondrial membrane integrity, permeability, and fluidity, thereby affecting the stability and activity of many IMM-related proteins, including those involved in the electron transport chain (ETC) and OXPHOS [6].

CL is a unique phospholipid that is localized and synthesized in the IMM and constitutes approximately 15–20% of the total mitochondrial phospholipids [7]. CL is unusual among all phospholipid species as it exhibits a dimeric structure with four acyl chains and two phosphatidyl moieties linked to glycerol. This unique structure of CL yields a conical shape, which is the origin of its curvature-sensing capability [8]. At the same time, CL can interact with a number of IMM proteins, including enzyme complexes that maintain ETC and ATP production. Furthermore, CL is sensitive to oxidative stress due to the presence of unsaturated fatty acyl chains. ROS-induced oxidation of CL affects the activity of the respiratory chain complex [9]. The accumulation of oxidized CL in the OMM leads to the opening of the mitochondrial permeability transition pore (mPTP) and the release of cytochrome c (Cyt C) from mitochondria into the cytoplasm [10], ultimately inducing apoptosis [11]. Thus, CL can play multiple roles, including the formation and stability of the mitochondrial cristae respiration supercomplex [12,13,14], mitochondrial quality control and dynamics [15], as well as mitochondrial biogenesis and protein import [16]. Notably, the roles of CL in regulating mitochondrial structure and functioning have been discussed in many excellent reviews [12,17,18,19,20,21,22,23,24,25]. Therefore, in this review, we will instead emphasize the recent advances in our understanding of its physiological roles in different mammalian cells and discuss the relationships of an abnormal CL content, composition, and oxidation with mitochondrial dysfunction and structural abnormality, as well as their implications in pathophysiological conditions and diseases.

## 2. CL Biosynthesis and Remodeling

Mature CL is a unique tetra-acylated diphosphatidylglycerolipid containing predominantly unsaturated fatty acids (FAs), which is produced through highly conserved synthetic and remodeling pathways. Briefly, the biosynthesis of CL begins with the conversion of phosphatidic acid (PA) to CDP-DAG (Figure 1), which can be catalyzed by CDS1/CDS2 in the ER [26] or by TAMM41 in the mitochondria [27]. Subsequently, CDP-DAG is transported to the IMM and then converted to phosphatidylglycerophosphate (PGP) by PGP synthase (PGS1) [28]. PGP is further dephosphorylated by protein tyrosine phosphatase 1 (PTPMT1) to generate phosphatidylglycerol (PG) [29]. On the stromal side of the IMM, CL is synthesized by CL synthase 1 (CLS1) through an irreversible condensation reaction of PG and CDP-DAG [19]. The newly synthesized CL is an immature CL characterized by variable-length and asymmetric saturated acyl chains around the glycerol head group [30].

To become mature CL, the newly synthesized CL undergoes remodeling, a process of deacylation and reacylation. CL is first deacylated to MLCL by phospholipase A2 (PNPLA8) [31,32], also called Ca^2+^-independent phospholipase A2 γ (iPLA2γ), which removes a saturated fatty acyl chain from CL [31], and then reacylated by acyltransferase to introduce an unsaturated fatty acyl chain [33]. Three enzymatic pathways have been identified to be involved in this reacylation reaction. The coenzyme-A-independent acyltransferase, TAZ, is located on the outer surface of the IMM and is a transacylase, performing acyl chain exchange between CL and phospholipids, mainly phosphatidylcholine (PC), to sequentially replace fatty acyl chains at all four acyl positions of the CL [34,35]. In addition to TAZ, ALCAT1 and MLCL AT1, located on the endoplasmic reticulum mitochondria-associated membrane and inner lobe of the IMM, respectively, are also able to catalyze CL remodeling by binding long polyunsaturated fatty acyl chains [33]. Both ALCAT1 and MLCL AT1 use the acyl chain of acyl-CoA to ensure the reacylation of MLCL [33,36,37,38]. It is noteworthy that the four fatty acyl chains of CL in metabolic tissues, including the heart, are dominated by linoleic acid (18:2), called tetralinoleoyl CL (TLCL), indicating that the enzymes involved in CL remodeling in such tissues may utilize linoleic acid as the major substrate [39]. However, ALCAT1 lacks a preference for linoleic acid as a substrate. Instead, CL remodeling by ALCAT1 results in the enrichment of docosahexaenoic acid (DHA) and arachidonic acids, leading to CL having a high sensitivity to oxidants [33]. As a consequence, increased expression of ALCAT1 may directly lead to oxidative stress and mitochondrial dysfunction in cells [33,39,40,41,42]. Actually, the diversity in the acyl chain composition in different cells and tissues is regulated by differences in CL synthesis, catabolism, and remodeling, and the acyl chain composition has been shown to play an important role in regulating inner membrane fluidity, structure, osmotic stability, and protein properties [43,44,45].

After de novo synthesis and remodeling, mature CL transfers and assembles in the IMM and OMM. The translocation of CL is mediated by three different enzymes, namely phospholipase scrambling, mitochondrial creatine kinase, and nucleoside polyphosphate kinase [46]. Interestingly, the activities of these kinases are also regulated by CL and require protein aggregation that promotes the formation of CL clusters and CL membrane domains [47]. These CL-enriched domains are thought to play an important role in the regulation of the mitochondrial membrane structure and morphology, as well as in the aggregation of proteins [47].

## 3. CL and Mitochondrial Cristae Formation

Mitochondrial cristae are folds of the mitochondrial inner membrane that provide an increase in surface area, enabling the surface area of the IMM to be more than four times that of the OMM [11]. Cristae are also functionally dynamic, with their shape and volume modulating the kinetics of chemical reactions and the structure of protein complexes [48]. CL and mitochondrial-shaping proteins are critical for the formation and maintenance of mitochondrial cristae. Structurally, the tetra-acyl chains of CL, together with its negatively charged polar moiety, give CL a conical shape [12]. The polar region is the top of the cone, and the flexible and variable acyl chains are the base of the cone, resulting in the tendency of CL to form a non-bilayer structure. Furthermore, the lateral area of the CL anion head is smaller than the lateral area of its four alkyl chains, and such an increased tapered shape causes the CL to exert lateral pressure in the bilayer membrane, causing negative curvature and the inner monolayer of the IMM to bend and form mitochondrial cristae [49]. In addition, CL spontaneously forms either inverse lamellar or hexagonal structures, depending on the saturation of the acyl chains and the pH of the solution. These structural features of CL are critical for maintaining the curvature of the mitochondrial cristae, the potential of the inner membrane, and the numerous protein interactions in and out of the mitochondria.

Abnormal CL biosynthesis and remodeling have been associated with altered cristae morphology and disrupted mitochondrial-shaping proteins, affecting the number, length, and organizing pattern of the cristae, and ultimately the shape of mitochondria and the progress of OXPHOS in both human patients and animal models [13,14,50,51,52,53,54,55,56,57,58]. Mitochondrial ATP synthase is a polymerase complex consisting of two functional domains: F0, located in the IMM, and F1, located in the mitochondrial matrix. ATP synthase has been shown to form higher oligomeric assemblies consisting of rows of dimers that are critical for cristae formation and stabilization [59]. CL appears to be critical for oligomerization and ordering in these ATP synthase assemblies, and an abnormal CL content has been shown to be associated with reduced oligomerization and ordering of ATP synthase [13,14,60,61,62]. On the other hand, the MICOS is the other machinery that has been identified to play a crucial role in maintaining the characteristic architecture of the IMM in addition to ATP synthase [63]. MICOS is a multi-subunit complex that has two core components, MIC10 and MIC60, and at least four additional IMM proteins. According to its two core components, MICOS can be organized into two subcomplexes: the Mic60-subcomplex, which is sufficient for crista junction formation, and the Mic10-subcomplex, which controls lamellar cristae biogenesis [64]. Importantly, MICOS components may interact with CL, and the interaction between CL and MICOS components may synergistically protect cristae structures [65,66]. Furthermore, a deficiency of CL biosynthesis and remodeling is also able to perturb the expression of MICOS proteins and assembly of the MICOS complex [66,67,68]. Therefore, the interaction of CL with mitochondrial-shaping proteins, together with its own biochemical profile, provides the basis for the formation and maintenance of mitochondrial cristae. It must be noted that CL deficiency in different cell types results in different morphological changes in mitochondrial cristae, ranging from swelling and reduction in number and length, to form bubble-like, onion-like, concentric, and highly interconnected structures [50,56,57,58]. The underlying cell-type-specific mechanisms, including the cell-type-specific changes in the MICOS complex and ATP5A synthase, are largely not well determined and remain to be further investigated.

## 4. CL Regulates Mitochondrial Function

It has been well-recognized that CL is crucial for regulating various mitochondrial functions, ranging from respiration and metabolism to apoptosis. First of all, CL may play a central role in regulating mitochondrial bioenergetics, not only because it is essential for cristae formation and maintenance, but also because it can bind and stabilize many enzymes of ETC [69]. Specific binding sites for CL have been observed in complexes I, III, and IV. CL binding may be required for the proper folding and optimal functioning of these ETC complexes, as abnormal CL binding to these complexes could cause subunit dissociation and a complete loss of activity, whereas CL reassociation leads to the stabilization of the quaternary structure and restoration of full activity [70,71]. CL can also regulate the formation of the ETC supercomplex. In the IMM, ETC complex I is organized with two complex III and multiple complex IV units to form a supercomplex, which ensures efficient substrate channeling between individual complexes [72,73]. Evidence from yeast to humans supports a role for CL in the higher-order organization of respiratory complexes. CL deficiency can cause instability in the respiratory supercomplex [74,75,76,77,78]. Thus, CL acts as a scaffold for the correct assembly of both individual respiratory complexes and supercomplexes, providing structural–functional connections between respiratory complexes, enabling the optimal transfer of electrons in the IMM, and preventing excessive ROS production [79]. In addition, CL may regulate mitochondrial energy production in other ways. CL plays a role in anchoring Cyt C to the IMM, which facilitates electron transfer from complex III to complex IV [80]. CL may also bind to mitochondrial ribosomes, thereby stabilizing the IMM binding of the protein translation machinery and supporting the biogenesis of mitochondrial OXPHOS proteins [81].

CL may also play an important role in regulating mitochondrial metabolism. The mitochondria is not only the powerhouse, but also one of the central organelles for metabolism, participating in numerous metabolic processes, including the citric acid cycle, FA oxidation, synthesis and degradation of amino acids, as well as synthesis of iron–sulfur clusters and heme [1]. It has been shown that CL may interact with a number of mitochondrial carrier proteins, including the ADP/ATP carrier (AAC) [82,83], phosphate carrier (PiC) [84], pyruvate carrier [85], tricarboxylate carrier [86], and carnitine/Acylcarnitine translocase, as well as participate in the regulation of acetyl-CoA synthesis [87], tricarboxylic acid (TCA) cycle [88], assembly of mitochondrial translocase and protein import [16], and ADP/ATP exchange [89].

It is important to note that CL is susceptible to peroxidative damage induced by ROS, as it contains a high amount of unsaturated acyl chains. In the mitochondria, complexes I and III are considered as the main sites for ROS generation [21]. Hence, its proximity to the site of ROS generation in the IMM further renders the oxidation of CL during defective supercomplex formation and activity [90,91]. Furthermore, Cyt C may also selectively oxidize CL, especially during apoptosis or in the presence of oxidants, such as H_2_O_2_ [92,93]. During apoptosis, the peroxidase activity of the Cyt C/CL complex targets the unsaturated acyl chain of CL, causing an increase in oxidized CL [93]. The oxidation of CL in turn affects mitochondrial functioning. First, the oxidation of CL has been shown to reduce the activity of complexes I, III, and IV and increase ROS production, which can be alleviated by CL supplementation [90], suggesting that CL oxidation decreases mitochondrial bioenergetics. Secondly, oxidized CL and its externalization on the OMM could induce mPTP opening and the mitochondrial release of Cyt C [94,95], and thus may play an important role in cell apoptosis. In particular, the oxidation of CL may facilitate the detachment of Cyt C from the IMM and its release from mitochondria to the cytoplasm, as Cyt C exhibits a lower affinity for oxidized CL [96]. The inhibition of CL oxidation in cells reduces Cyt C release [97].

In addition to Cyt C, CL binds to many other apoptotic proteins and has been proposed to participate in different steps of the apoptotic process. CL transferred from the IMM to the OMM may be a signal for the binding of apoptotic proteins [98]. Following Fas receptor activation, pro-Caspase 8 migrates to the CL-enriched regions of the OMM [99]. The activation of Caspase-8 results in the cleavage of the pro-apoptotic protein Bid to the active truncated Bid (tBid), which is then translocated to the OMM. The binding of tBid to CL could further increase the transfer of CL to the OMM, generating more negative charges in the OMM and recruiting more polycationic apoptotic proteins to the mitochondria [100]. Evidence also shows that the recruitment and oligomerization of Bak-Bax in the OMM is a CL-dependent process [101]. Clearly, perturbations in the CL distribution, metabolism, and redox status may play an important role in both the induction and progression of cell apoptosis, mainly via the CL–protein interaction. However, how the lipid microenvironment is initiated and regulated, as well as how the CL–protein interaction is temporally and spatially regulated, are still not well characterized and remain to be further investigated.

## 5. CL Deficiency and Diseases

Deficiencies in CL biosynthesis and remodeling may lead to abnormal mitochondrial ultrastructures and functioning. Meanwhile, the oxidation of CL in unsaturated acyl chains upon oxidative stress changes its own biophysical properties, disrupts binding with other proteins, and may eventually perturb mitochondrial functions. CL abnormalities, including changes in the CL content, FA acyl chain composition, and oxidation, have been highlighted in a range of human diseases, including Barth syndrome, heart failure, aging, metabolic disorders, and neurodegenerative diseases [18,21,102]. Here, we summarize recent studies, mostly from human patients and genetically engineered mouse models (Table 1), and discuss how these studies promote our understanding of the relationship between CL abnormalities and diseases.

## 6. CL and Cardiomyopathy

Cardiomyopathy is a disease of the heart muscle that makes it harder to pump blood to the rest of the body, which can eventually lead to heart failure [124,125]. CL deficiency has long been known to be a direct cause of heart defects due to the huge number of mitochondria in cardiac cells and the essential role of CL in regulating mitochondrial structure and functioning. First of all, changes in the CL content, CL species composition, and CL biosynthesis and remodeling enzymes have been observed in diseased and aging hearts [126,127,128], which may further contribute to progressive changes in the mitochondrial functioning and energy metabolism in cardiac disease. A loss of 20–25% of the CL amount was found in the ischemic hearts, and intriguingly, such a reduction in the CL levels appears to precede losses in other phospholipids [129]. A reduced total CL content and changes in CL species composition were detected in both aging human hearts [126,127] and patients with heart failure [130]. Changes in the CL pool have also been demonstrated in rat models of heart failure induced by both pressure overload and spontaneous hypertension [131,132]. Evidently, normal CL biosynthesis and remodeling are required for normal cardiac development and physiology. Our recent study using a cardiac-specific PTPMT1 mouse model demonstrated that deficiency in CL biosynthesis induced by PTPMT1 deletion in mouse cardiomyocytes decreased embryonic cardiac cell proliferation, resulting in decreased thickening of ventricular walls, arrestment of heart development and growth, and eventually embryonic lethality [13]. Furthermore, various cardiac defects, including dilated cardiomyopathy, hypertrophic cardiomyopathy, left ventricular myocardial noncompaction, and ventricular arrhythmia, have been observed in patients with Barth syndrome, a rare X-linked genetic disorder caused by mutations in TAZ [17,96,133,134]. Such cardiac phenotypes can be partially recapitulated in genetically engineered mouse models with knockdown [52,56,58,107], germline deletion [113], or cardiac-cell-specific knockout of the mouse TAZ gene [14,51,113]. On top of baseline phenotypes, TAZ-deficient mice also exhibit higher susceptibility to a lipotoxic hypertrophic cardiomyopathy in response to a high-fat diet [108]. CL deficiency has also been linked to cardiomyopathy observed in patients with dilated cardiomyopathy with ataxia (DCMA) caused by a mutation in the gene encoding for mitochondrial protein DNAJC19 [135], as well as in patients with Sengers syndrome caused by mutations in the gene encoding for the mitochondrial acylglycerol kinase [136].

On the other hand, studies have also been performed to investigate whether the restoration of the CL content or targeting CL biosynthesis and remodeling could alleviate mitochondrial abnormalities and improve cardiac performance. In fact, it has been shown that the deletion of ALCAT1 is able to prevent the onset of T4-induced cardiomyopathy and cardiac dysfunction [41]. Furthermore, the ablation of ALCAT1 by gene deletion or pharmacological inhibition can alleviate both cardiac and kidney injury following myocardial infarction [39,120,121]. Similarly, the deletion of PNPLA8 in adult mouse cardiomyocytes is also protective from myocardial ischemia/reperfusion injury [118], although the global deletion of PNPLA8 in mice leads to various physiological defects, including growth retardation, kyphosis, muscle weakness, and glomerular injury [115,116,117]. In addition, the application of exogenous CL-containing liposomes to mitochondria isolated from ischemic rat hearts can also restore the activity of ETC complexes to normal levels [137,138]. The mitochondrial-targeting peptide SS-31 has been shown to protect CL from oxidation by cytochrome c peroxidase and reduce the infarct size in ischemia/reperfusion [139,140].

## 7. CL in Neuronal Diseases

Neurons, like cardiomyocytes, also require a high metabolic rate, with the brain consuming over 20% of the total body energy, which is ensured by both glucose metabolism and OXPHOS. Intriguingly, the brain displays a different acyl chain profile that is enriched by long-chained FA (20:4 and 22:6) compared with cardiac tissues, which display a much more homogenous acyl chain pattern predominantly incorporating linoleic acid (18:2) [141]. Indeed, CL plays an important role in both cardiac and neuronal cells, and in the past two decades, numerous clinical and experimental studies from either patients or animal models have provided evidence that aberrant CL metabolism may be linked to neurological dysfunction [18]. First, CL biosynthesis and remodeling are required for normal brain development. The deletion of PTPMT1 or CRLS1 in mouse neural precursor/stem cells or specific types of neurons demonstrated that a deficiency of CL biosynthesis may disrupt mitochondrial functioning and structure in neurons, arresting cell cycles and inducing the loss of neurons. This further results in various brain abnormalities, ranging from moderate ataxia, growth retardation, and compromised cerebral development to totally blocked cerebellar development and premature lethality [50,105]. It has been shown that TAZ knockdown mice develop cognitive deficiencies and hippocampal alteration [109], indicating that CL deficiency may also affect human neurocognitive development, which is further highlighted by the evidence that Barth syndrome patients may develop various learning and behavior deficits [142,143,144,145]. The deletion of PNPLA8 also causes deficiencies in spatial learning and memory performance [32]. Secondly, CL abnormalities represent a molecular hallmark of aging and neurological disorders. A lower CL content has been found in brain tissues collected from murine models with aging [146,147,148,149], Alzheimer’s disease (AD) [150], Amyotrophic lateral sclerosis (ALS) [151], and traumatic brain injury (TBI) [152,153], as well as from TBI patients [154]. In addition to decreasing CL levels, CL peroxidation or changes in the CL species composition have also been observed in brain tissues from murine models of Parkinson’s disease [155,156] and from patients with TBI [154]. It is important to note that changes in the CL levels can also be found in serum from patients with frontotemporal dementia [157] and animal models with PD and TBI [156,158], suggesting a potential role for CL as a biomarker of neurological diseases. However, it remains largely unclear if the decrease in the CL levels and peroxidation of CL have causal implications in neuronal damage and aging, or if it is just a part of a mitochondrial adaptive response triggered by aging and stress. Inducible deletion of PTPMT1 in cerebella only causes a transient defect in adult mice that soon recover and appear grossly normal afterward [105], indicating that CL might play a less important role in the maintenance of brain functioning than in brain development. Intriguingly, the ablation of ALCAT1 activity in mice could either attenuate motor neuron dysfunction, inflammation, and muscle atrophy in a model of ALS [122], or inhibit MPTP-induced neurotoxicity, apoptosis, and motor deficits [40], indicating that ALCAT1 may be a therapeutic target for treating neuronal diseases. Regardless, the role of CL in neurons is not fully understood, and more studies should be performed in the future to investigate the effects of CL deficiency in different types of neurons and in distinct neurological disease models.

## 8. CL in Immune Cells

It has been shown that mitochondria in immune cells play a pivotal role in regulating cell development, activation, proliferation, differentiation, and death. In different types of immune cells, mitochondria undergo distinct adaptions in their morphology, ultrastructure, function, and metabolism in response to environmental stimulation [159,160]. Interestingly, the content and the acyl chain composition of CL in lymphocytes also undergo changes accordingly upon stimulation [103,114]. Certainly, CL may also function as a key player in regulating mitochondrial functioning and structure in immune cells, and, consequently, their cell fate and fitness as well. First, it has been shown that normal CL biosynthesis is essential for hematopoietic stem cell differentiation [104]. The deletion of PTPMT1 in the hematopoietic cells causes a complete block of stem cell differentiation, evidenced by the loss of hematopoietic progenitors, and eventually results in severe anemia and premature lethality in mice. PTPMT1 knockout stem cells also fail to differentiate into progenitor cells in colony-forming unit assays or give rise to a blood cell lineage when cocultured with OP9 stromal cells [104]. Surprisingly, the deletion of PTPMT1 from lineage progenitors using LysM-Cre (granulocyte and macrophage progenitors), LCK-Cre (T cell progenitors), and CD19-Cre (B cell progenitors) does not disturb the cell development of each specified lineage or cause gross abnormality in mice [104], suggesting that PTPMT1-mediated CL biosynthesis plays a vital role in hematopoietic stem cells, but is dispensable for late lineage progenitors. However, a recent study using CD4-Cre, another T-cell-specific Cre, to delete PTPMT1 in mice showed that a deficiency of CL biosynthesis may significantly perturb the development, metabolism, and function of memory CD8+ T Cells, as indicated by a decrease in the CD8+ T cell numbers and cytokine production, in turn enhancing CL synthesis, which can promote the generation of memory-like CD8+ T Cells [103]. In addition, this study also provided evidence that a reduced number and impaired function of CD8+ T cells can be observed in both TAZ KO mice and Barth syndrome (BTHS) patients following long-term systemic CL deficiency [103]. Furthermore, TAZ-mediated CL remodeling is also essential for the activation of intraepithelial lymphocytes (IELs), a type of epithelial-resident T lymphocyte located at the intestinal barrier, which can offer swift protection against invading pathogens. TAZ deficiency can significantly reduce IEL activity and increase the risk of immunopathology, as evidenced by decreased cell proliferation in response to anti-CD3 stimulation and an elevated parasite load after infection by a small intestinal parasite [114]. On the other hand, evidence also suggests that CL may play a role in regulating B cell functioning and development. TAZ deficiency has been shown to reduce the cell proliferation of LPS-stimulated B cells [111], and hypogammaglobulinaemia and B cell lymphopaenia were recently observed in a BTHS patient carrying a novel TAZ mutation [161]. However, future studies utilizing genetically engineered animal models should be performed to confirm whether and how CL regulates B cell development and function.

It is noteworthy that most patients with Barth Syndrome develop mild to severe intermittent or persistent neutropenia, which is most often associated with mouth ulcers, painful gums, pneumonia, and sepsis, due to possible bacterial infection [144,162,163]. Reduced circulating neutrophil numbers have also consistently been found in adult TAZ knockout mice [113]. Neutrophils are the predominant cell type among leukocytes, but have the shortest life span, being eliminated by tissue macrophages within a few days after leaving the bone marrow [164,165]. It remains largely unclear whether neutropenia in BTHS patients or in TAZ knockout mice is due to diminished production or increased clearance of mature neutrophils. In fact, directed motility and killing activity in neutrophils collected from BTHS patients are comparable to those of healthy donors [166]. Although many circulating BTHS neutrophils exhibit binding with annexin V, these cells do not show other markers of apoptosis and are not phagocytosed by macrophages [166]. A recent study utilizing an ex vivo method also suggests that the deletion of TAZ in mice does not affect neutrophil development and functioning, including phagocytosis and the production of cytokines and ROS [167]. Interestingly, TAZ-deficient neutrophils appear to be more sensitive to ER-stress-induced apoptosis [167], suggesting that it may be worth investigating whether environmental, nutritional, or infectious stress increase the susceptibility of BTHS neutrophils to apoptosis and affect the number of circulating cells in future studies.

## 9. CL and Metabolic Disorders

Changes in the CL content and FA composition have been observed in various animal models of metabolic disorders. In diabetic myocardium after streptozotocin induction, the CL content has been found to be dramatically reduced, which is accompanied by a significant increase in FA (22:6)-containing CL molecular species [168]. In obese diabetic (ob/ob) mice, a similar redistribution from FA (18:2)-containing CL molecular species to FA (22:6)-containing CL molecular species can also be observed [168]. On the other hand, CL biosynthesis and remodeling may play a direct role in regulating systemic metabolism and energy homeostasis. It has been shown that TAZ deficiency can reduce basal insulin secretion and increase fibrosis in pancreatic islets [112], suggesting that CL may regulate β cell functioning and survival. In adipose tissue, lipid metabolism pathways can be significantly activated during cold adaption, which is further highlighted by the upregulation of PG and CL, especially newly synthesized CL with shorter and more saturated acyl chains, in brown and beige adipose tissues [55]. Indeed, CL biosynthesis plays an essential role in maintaining energy homeostasis and insulin sensitivity. The deletion of CRLS1 in mouse brown and beige fat may abolish adipose thermogenesis and glucose uptake, resulting in lower cold tolerance and decreased insulin sensitivity [55]. In addition to adipose tissues, CL biosynthesis in hepatocytes also plays an important role in regulating insulin sensitivity. Furthermore, the deletion of CRLS1 in mouse hepatocytes increases inflammatory responses and fibrosis, and significantly exacerbates high-fat-diet-induced insulin resistance and hepatic steatosis [106]. It is noteworthy that CL remodeling pathways may also play an important or even a reverse role in regulating insulin sensitivity. It has been shown that TAZ deficiency could prevent hepatic steatosis and high-fat-diet-induced obesity [110]. Moreover, the deletion of ALCAT1 or pharmacological inhibition of ALCAT1 in mice has been shown to reduce high-fat-diet-induced obesity and insulin resistance, attenuate hepatic lipogenesis and fibrosis, and thus prevent the onset of diet-induced nonalcoholic fatty liver diseases [33,123]. The deletion of PNPLA8 in mouse hepatocytes also enhances glucose clearance and reduces FA accumulation in the liver after high-fat-diet stimulation [119]. Taken together, all these studies strongly demonstrate the critical role of CL biosynthesis and remodeling in maintaining metabolic functions and in regulating the development of metabolic disorders.

## 10. Conclusions

In the past few decades, clinical and experimental studies from human and animal models, especially from genetically engineered mouse models, have demonstrated that CL plays an essential role in maintaining mitochondrial ultrastructure and functioning in multiple cell types. Aberrant CL metabolism has been directly linked to a number of developmental and physiological abnormalities, including embryonic lethality, cardiomyopathy, failure of HSC differentiation, immune cell defects, neuronal dysfunction, and metabolic disorders. However, the physiological roles of CL in many cell types and how CL deficiency in these cells contributes to the pathophysiology of certain diseases still remain to be determined; thus, more studies could be conducted in the future to investigate the cell-type-specific role of CL biosynthesis and remodeling, and to explore potential approaches to restore CL abnormalities. Although substantial advances in lipidomic techniques have significantly promoted our understanding of the acyl chain composition of CL and its related metabolites, the pathways and underlying molecular mechanisms involved in regulating CL metabolism in both physiological and pathophysiological conditions remain largely unclear. In summary, CL indeed contributes to numerous aspects of cellular functions and human health, but the present studies are not sufficient for us to fully understand this unique phospholipid with four acyl chains. Continuous investigation incorporating innovative and multi-disciplinary approaches will be crucial to further characterizing what CL does inside the cell and how.

## Figures and Tables

**Figure 1 genes-13-01889-f001:**
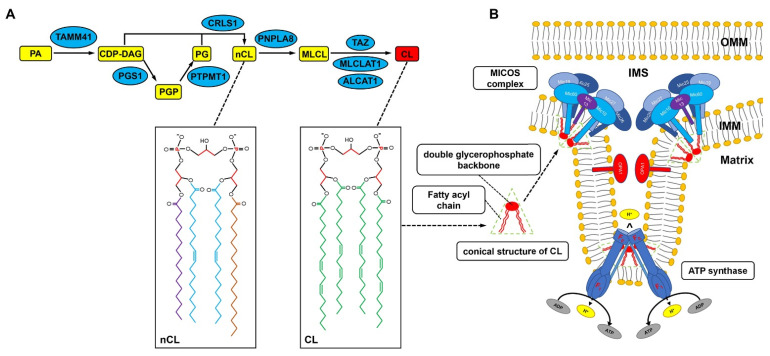
Schematic diagram depicting the synthesis, remodeling, and localization of CL inside the mitochondria. (**A**) Inside the mitochondria of mammalian cells, the biosynthesis of CL begins with the conversion of phosphatidic acid (PA) to CDP-DAG catalyzed by TAMM41, followed by the transfer of a phosphatidyl group from CDP-DAG to glycerol-3-phosphate to form PGP, a process catalyzed by PGS1. PGP is subsequently dephosphorylated by PTPMT1 to generate PG, and CL is synthesized by CRLS1 through an irreversible condensation reaction of PG and CDP-DAG. The newly synthesized CL (nCL) contains a mixture of fatty acyl chains differing in length and saturation. To become mature CL, nCL undergoes a series of deacylations and reacylations, a process called remodeling. nCL is first deacylated to monolysocardiolipin (MLCL) by phospholipase A2 PNPLA8 and then reacylated by acyltransferases, including tafazzin (TAZ), acyl-CoA: lysocardiolipin acyltransferase 1 (ALCAT1), and MLCL acyltransferase-1 (MLCL AT1) to achieve a final symmetric acyl composition in mature CL. (**B**) The tetra-acyl chains and the negatively charged polar moiety give CL a conical shape, which results in CL causing negative curvature in the mitochondrial membrane and the IMM bending, forming the cristae. In addition, the mitochondrial contact site and cristae-organizing system (MICOS) complex and ATP synthase are also essential for the formation of mitochondrial cristae.

**Table 1 genes-13-01889-t001:** Genetically engineered mouse models of Barth syndrome and CL-deficiency-related diseases. ^1^ KO, knockout; ^2^ KD, knockdown.

Genes	Mouse Models	Major Phenotypes	Ref.
*Ptpmt1*	Global KO ^1^	Embryonic lethality before E8.5	[29]
	Cardiac-specific KO by Tnt-Cre	Abnormal heart development, embryonic lethality at around E16.5, mitochondrial cristae malformation, and mitochondrial dysfunction	[13]
	Induced cardiac-specific KO in adult mice by αMHC-MerCreMer	No obvious baseline phenotype	
	T-cell-specific KO by CD4-Cre	Reduced CD8+ cell numbers and abnormal memory CD8+ T cell function	[103]
	HSC-specific KO by Vav1-Cre	Failure of HSC differentiation, pancytopenia, anemia, and premature lethality	[104]
	Granulocyte-macrophage progenitors-specific KO by LysM-Cre	No obvious baseline phenotype	
	T-cell-specific KO by LCK-Cre	No obvious baseline phenotype	
	B-cell-specific KO by CD19-Cre	No obvious baseline phenotype	
	Neural cell-specific KO by Nestin-Cre	Cell cycle arrest of neuronal progenitors, growth retardation, ataxia, and premature lethality	[105]
	Purkinje cell-specific KO by PCP2-Cre	Minimal structural changes in cerebellum and altered walking gaits	
	Granule cell-specific KO by Atoh1-Cre	Abnormal anterior lobules and altered walking gaits	
	Dual neural-cell-specific KO by PCP2-Cre and Atoh1-Cre	Abnormal anterior lobules and altered walking gaits	
	Induced KO in adult mice by CAG-CreER	Transient defects, including ataxia, tremor, and impaired motor coordination, but soon recovered	
*Crls1*	Global KO	Early embryonic lethality at the peri-implantation stage	[50]
	Neuron-specific KO by Camk2α-Cre	Neuronal loss and gliosis in the forebrain, and lethality at age 12–14 months	
	Adipocyte-specific KO by Adipoq-Cre	Less adipose tissue and reduced cold tolerance and insulin sensitivity	[55]
	Induced brown and beige adipose-specific KO by UCP1-CreER	Paler brown fat, insulin resistance	
	Hepatocyte-specific KO by Alb-Cre	Exacerbated insulin resistance and hepatic steatosis induced by high-fat diet, and aggravated inflammatory response and fibrosis induced by a high-fat and high-cholesterol diet	[106]
*Taz*	Doxycycline-induced global shRNA KD ^2^	Prenatal and perinatal lethality, embryonic diastolic dysfunction, and myocardial noncompaction	[56]
	Doxycycline-induced global shRNA KD	Decreased body weight and left ventricular dilation and dysfunction at an age of 8 months	[58]
		Impaired skeletal muscle force generation at an age of 2 months and reduced ejection fraction at an age of 7–10 months	[107]
		Increased cardiac lipotoxicity due to high-fat diet	[108]
		Cognitive deficiency and hippocampal alteration	[109]
		Resistant to high-fat-induced obesity, insulin resistance, and hepatic steatosis	[110]
		Reduced cell proliferation of LPS-stimulated B cells	[111]
		Reduced plasma insulin, impaired insulin secretion under low-glucose conditions, and increased fibrosis in pancreatic islets	[112]
		Heart failure with a preserved ejection fraction and age-dependent progression of diastolic dysfunction, and prolonged QRS duration	[52]
	Global KO	20% die prenatally, survivors develop neutropenia, premature lethality, growth retardation, skeletal myopathy, and heart failure	[113]
		Reduced number and impaired function of CD8+ T cells	[104]
		Bone marrow KO chimeras and reduced activities of intraepithelial lymphocytes in response to anti-CD3 and parasite infection	[114]
	Cardiac-specific KO by αMHC-Cre	Progressive dilated cardiomyopathy and cardiac fibrosis without fetal and perinatal lethality	[113]
		Increased vulnerability to arrhythmia	[51]
	Cardiac-specific KO by Xmlc2-Cre	No embryonic lethality, ~5% die before an age of 2 months, while survivors exhibit ventricular dilation and contractile dysfunction	[14]
*PNPLA8*	Global KO	Growth retardation, cold intolerance, reduced exercise endurance, and greatly increased mortality from cardiac stress after transverse aortic constriction	[115]
		Deficiency in spatial learning and memory performance	[32]
		Loss of podocytes in aging mice; higher albuminuria and more podocyte injury and loss in response to nephritis	[116]
	Global KO	Growth retardation, kyphosis, and muscle weakness	[117]
	Induced cardiac-specific KO in adult mice by αMHC-MerCreMer	Protective from myocardial ischemia/reperfusion injury	[118]
	Hepatocyte-specific KO by MMAP-Cre	Enhances glucose clearance and reduces FA accumulation after high-fat diet	[119]
*ALCAT1*	Global KO	Impairs kidney injury after myocardial infarction	[120]
		Improves cardiac performance after myocardial infarction	[121]
	Global KO	Inhibits MPTP-induced neurotoxicity, apoptosis, and motor deficits	[40]
		Mitigates hyperthyroid cardiomyopathy and ventricular fibrosis	[41]
		Reduced high-fat-diet-induced obesity and insulin resistance	[33]
		Mitigates CHD and its related pathogenesis, including dilated cardiomyopathy, left ventricle dysfunction, inflammation, fibrosis, and apoptosis	[39]
		Attenuates motor neuron dysfunction, neuronal inflammation, and skeletal muscle atrophy in SOD1G93A mice	[122]
		Attenuates hepatic lipogenesis and fibrosis and prevents the onset of diet-induced nonalcoholic fatty liver diseases	[123]

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
