# Peer review of "Cardiolipin Regulates Mitochondrial Ultrastructure and Function in Mammalian Cells"

_genes, 2022, doi:10.3390/genes13101889_

Round 1

Reviewer 1 Report

This is a very interesting and very well written review on a very important subject of biology with numerous pathophysiological implications.

As stated by the authors (lines 57-59), a number of reviews already appeared on the subject of cardiolipin (CL) [12, 17-25]. However, analysis of the corresponding titles indicates that the most recent ones are centered on only one aspect of CL: heart [17] (2022), [22] (2020); nervous system [18] (2021); immune system [19] (2020). On the other hand, the most recent of global reviews [12,20,21,23-25] is from 2019 [21]. So, I fully agreee with the claim by the authors (lines 57-63) that their review will provide the most recent advances on the field.

I have only minor suggestions to improve the manuscript:

1. In several places, the authors report previous studies showing that ALCAT1 displays deleterious effects. These concern heart and kidney (lines 273-275 [112-115], nervous system (lines 318-321 [144,145] and metabolism (lines 397-400 [33,162]. A priori, this appears rather paradoxical for an acyltransferase expected to maintain CL integrity and to avoid accumulation of MLCL. I suggest the authors discuss those data in term of hypothetical mechanisms (apparently a relationship with lipid oxidation). This could be done in the conclusion section.

2. Immune system: one paper centered on intraepithelial resident T-lymphocytes (IELs) should be included (Konjar et al, Sci Immunol 2018, 3, eaan2543). In particular, authors show that resting IELs are rich in polyunsaturated CL species, which are decreased upon activation. They provide convincing evidence that these changes in CL composition are related to IEL metabolism and activation state. Interestingly, in lines 286-289, the present review relates similar differences between brain and heart CL. This would deserve mention and maybe some speculative discussion.

3. Abbreviations should be defined the first time they appear in the text. This should be corrected for MLCL (line 86 instead of line 93), MICOS (line 90 instead of line 143), ALCAT1 (line 87 instead of line 101), MLCLAT1 (line 87 instead of line 102), TAZ (line 87 instead of line 98), BTHS (line 349, not defined)

4. line 95: gamma symbol is missing fot iPLA2

5. line 269: correct "to investigate"

6. Table 1 is never mentioned in the text. This could be done in the Conclusion section, announcing that very complete analysis just to end the review.

7. References are not presented according to rules given to authors by MDPI journals.

Author Response

We thank the Reviewer for the favorable comments that “This is a very interesting and very well written review on a very important subject of biology with numerous pathophysiological implications.

Specific Comments:

  1. In several places, the authors report previous studies showing that ALCAT1 displays deleterious effects. These concern heart and kidney (lines 273-275 [112-115], nervous system (lines 318-321 [144,145] and metabolism (lines 397-400 [33,162]. A priori, this appears rather paradoxical for an acyltransferase expected to maintain CL integrity and to avoid accumulation of MLCL. I suggest the authors discuss those data in term of hypothetical mechanisms (apparently a relationship with lipid oxidation). This could be done in the conclusion section.

We thank the Reviewer for this excellent comment. We also totally agree with the Reviewer that TAZ- and ALCAT1-mediated CL remodeling may play a different physiological role in mammalian cells. According to the Reviewer’s suggestion, we have added the following sentences in the section 2 (CL Biosynthesis and Remodeling) of our revised manuscript (Page 3): “It is noteworthy that the four fatty acyl chains of CL in metabolic tissues including the heart are dominated by linoleic acid (18:2), called tetralinoleoyl CL (TLCL), indicating that the enzymes involved in CL remodeling in such tissues may utilize linoleic acid as the major substrate [39]. However, ALCAT1 lacks a preference for linoleic acid as a substrate. Instead, CL remodeling by ALCAT1 results in enrichments of docosahexaenoic acid (DHA) and arachidonic acids, leading to a high sensitivity of CL to oxidants [33]. As a consequence, increased expression of ALCAT1 may directly lead to oxidative stress and mitochondrial dysfunction in cells [33,39-42].”  

  1. Immune system: one paper centered on intraepithelial resident T-lymphocytes (IELs) should be included (Konjar et al, Sci Immunol 2018, 3, eaan2543). In particular, authors show that resting IELs are rich in polyunsaturated CL species, which are decreased upon activation. They provide convincing evidence that these changes in CL composition are related to IEL metabolism and activation state. Interestingly, in lines 286-289, the present review relates similar differences between brain and heart CL. This would deserve mention and maybe some speculative discussion.

We thank the Reviewer for pointing out that we should include a new reference. As suggested by the Reviewer, we have included this reference in the revised manuscript. We also added the following sentences in the section 8 (CL in immune cells) of our revised manuscript (Page 10): “Furthermore, TAZ-mediated CL remodeling is also essential for the activation of intraepithelial lymphocytes (IELs), a type of epithelial-resident T lymphocytes located at the intestinal barrier, which can offer swift protection against invading pathogens. TAZ deficiency can significantly reduce IEL activity and increase the risk of immunopathology, evidenced by decreased cell proliferation in response to anti-CD3 stimulation and elevated parasite load after the infection of a small intestinal parasite [114].”

We also added the mouse model information in the Table 1 in the revised manuscript.

We also agree with the Reviewer that it is very interesting that CL in heart, brain, and immune cells have different acyl chain composition. Actually, CL species are key modulators of mitochondrial proteins, and their composition varies between cell types, which may be due to differences in CL synthesis, catabolism, and remodeling. As suggested by the Reviewer, we have now added the following sentence in the section 8 (CL in immune cells) of our revised manuscript to read (Page 10): “Interestingly, the content and the acyl chain composition of CL in lymphocytes also undergoes changes accordingly upon the stimulation [103,114].”

We also added the following sentence in the section 2 (CL Biosynthesis and Remodeling) of our revised manuscript (Page 3): “Actually, the diversity in the acyl chain composition in different cells and tissues is regulated by differences in CL synthesis, catabolism, and remodeling, and the acyl chain composition has been shown to play an important role in regulating inner membrane fluidity, structure, osmotic stability, and protein properties [43-45].”

  1. Abbreviations should be defined the first time they appear in the text. This should be corrected for MLCL (line 86 instead of line 93), MICOS (line 90 instead of line 143), ALCAT1 (line 87 instead of line 101), MLCLAT1 (line 87 instead of line 102), TAZ (line 87 instead of line 98), BTHS (line 349, not defined)

We have corrected these mistakes in the revised manuscript.

  1. line 95: gamma symbol is missing fot iPLA2

iPLA2” has been corrected to “iPLA2γ in the revised manuscript.

  1. line 269: correct "to investigate"

We have corrected this in the revised manuscript.

  1. Table 1 is never mentioned in the text. This could be done in the Conclusion section, announcing that very complete analysis just to end the review.

We apologized for this mistake. We have corrected this in the revised manuscript.

  1. References are not presented according to rules given to authors by MDPI journals.

We have corrected the reference style according to the MDPI journal’s rule. 

Reviewer 2 Report

I think this is an important review of the recent papers on the regulation of mitochondrial construction and function by CL.

I would like to make one minor comment.

#1 I am very interested in your detailed study of Table 1, and it is very informative, but it seems a little confusing. Would it be difficult to make it in such a way that it is classified by phenotype next to genotype? I would appreciate your consideration.

Author Response

We thank the Reviewer for the positive comments that “This is an important review of the recent papers on the regulation of mitochondrial construction and function by CL.

Specific comments:

#1 I am very interested in your detailed study of Table 1, and it is very informative, but it seems a little confusing. Would it be difficult to make it in such a way that it is classified by phenotype next to genotype? I would appreciate your consideration.

We appreciate the Reviewer for agreeing that “The Table 1 is very informative”. We also thank the Reviewer for the comment to reorganize the table by classifying the phenotype next to genotype. However, it seems to be difficult for us. The main problem is that the phenotypes described in different models are not exactly the same even in the same tissue. We now bold the interval lines between the different genotypes in the table to increase the readability.